# An Overview of Pentatricopeptide Repeat (PPR) Proteins in the Moss *Physcomitrium patens* and Their Role in Organellar Gene Expression

**DOI:** 10.3390/plants11172279

**Published:** 2022-08-31

**Authors:** Mamoru Sugita

**Affiliations:** Graduate School of Informatics, Nagoya University, Chikusa-ku, Nagoya 464-8601, Japan; sugita@gene.nagoya-u.ac.jp; Tel.: +81-52-789-5376

**Keywords:** pentatricopeptide repeat (PPR) protein, moss, *Physcomitrium patens*, plastids, mitochondria, post-transcriptional regulation, RNA processing, RNA splicing, RNA editing, RNA stabilization

## Abstract

Pentatricopeptide repeat (PPR) proteins are one type of helical repeat protein that are widespread in eukaryotes. In particular, there are several hundred PPR members in flowering plants. The majority of PPR proteins are localized in the plastids and mitochondria, where they play a crucial role in various aspects of RNA metabolism at the post-transcriptional and translational steps during gene expression. Among the early land plants, the moss *Physcomitrium* (formerly *Physcomitrella*) *patens* has at least 107 PPR protein-encoding genes, but most of their functions remain unclear. To elucidate the functions of PPR proteins, a reverse-genetics approach has been applied to *P. patens*. To date, the molecular functions of 22 PPR proteins were identified as essential factors required for either mRNA processing and stabilization, RNA splicing, or RNA editing. This review examines the *P. patens* PPR gene family and their current functional characterization. Similarities and a diversity of functions of PPR proteins between *P. patens* and flowering plants and their roles in the post-transcriptional regulation of organellar gene expression are discussed.

## 1. Introduction

Plastids and mitochondria, which are intracellular organelles in plants and algae, evolved from ancestral prokaryotic organisms, cyanobacteria and α-proteobacteria, respectively [1]. Chloroplasts (one type of plastid) carry out oxygen-evolving photosynthesis while mitochondria perform aerobic respiration, processes that are essential for almost all cellular functions [1]. Plastids and mitochondria have retained their own genome and transcription-translation machinery, which is composed of nuclear- and organellar-encoded proteins and RNAs [2,3]. Most organellar genes are organized in clusters and are often co-transcribed as polycistronic precursor RNAs that are then post-transcriptionally processed into multiple intermediate or mature RNAs [4,5,6,7]. Precursor, intermediate and mature RNAs are relative stable and accumulate at respective steady-state levels. Such post-transcriptional RNA processing, including RNA splicing, RNA cleavage and translational initiation, represents an important step in the control of organellar gene expression (Figure 1). The stabilization of RNA is also important for efficient translation. This complex processing is accomplished mostly by nuclear-encoded RNA-binding proteins such as pentatricopeptide repeat (PPR) proteins [8,9,10].

The large PPR gene family was first characterized in 2000 by the presence of tandem arrays of a degenerate 35-amino-acid (aa) repeat, which folds into a pair of α-helices and is widely distributed in fungi, animals and plants [11]. The same gene family, named the plant combinatorial and modular protein or PCMP family, was independently reported as a novel family that was unique to plants by Aubourg et al. [12]. They are categorized into a helical repeat protein family, including Pumilio (Puf repeat), β-catenin (Arm repeat) and pp2A (HEAT repeat) [13]. Plant PPR proteins are grouped into two subfamilies, P and PLS. The P subfamily (or P type) consists of canonical 35 aa PPR (P) motifs while the PLS subfamily is organized by repeated units of the P motif, long PPR (L) and short PPR (S) motif variants [14]. In the original definition by Lurin et al. [14], the PLS subfamily was further divided into four types, PLS, E, E+ and DYW, based on their PPR motif composition and characteristic C-terminal domain structures. Thereafter, based on a structural modeling approach, Cheng et al. [15] defined 10 different variants of the PPR motif (P, P1, P2, L1, L2, S1, S2, SS, E1 and E2) and divided the PPR proteins into six types, P, PLS, E1, E2, E+ and DYW (Figure 2). The DYW domain, based on the original definition, was composed of 106 aa [14] while the DYW domain, based on the new definition, becomes expanded to about 136 aa. Unlike vascular plants, only three types (P, PLS and DYW) are found in the moss *Physcomitrium patens*.

Nucleus-encoded PPR proteins constitute an extraordinarily large family in land plants, comprising 500 members in *Arabidopsis thaliana* (Arabidopsis) and 5,000 members in the whisky fern, *Selaginella moellendorffii* [15]. The majority of plant PPR proteins are localized in plastids or mitochondria [14,16] where they play important roles in a wide range of physiological and developmental functions [8,9,10]. Using forward genetics, various mutants that are defective in photosynthesis, respiration, cytoplasmic male sterility or embryogenesis were isolated and many *PPR* genes were then identified as being responsible for the mutated phenotypes [8,9,10,14]. Most PPR proteins that have been investigated are required for various post-transcriptional steps that are associated with RNA in plant organelles [17,18,19,20,21,22,23,24]. Thus, in the early 2000s, functional analyses of PPR proteins were performed using flowering plants, namely Arabidopsis, rice and maize. In contrast to studies performed in flowering plants, knowledge regarding PPR proteins in early land plants was limited [25]. In this review, the current knowledge of the function of *P. patens* PPR proteins in plastids and mitochondria is summarized, and attempts are made to highlight the differences and similarities of PPR proteins between mosses and angiosperms.

## 2. *P. patens* Is a Model Plant for Studying the Molecular Function of *PPR* Genes

The past two decades, my colleagues and I have engaged in the study of the functional analyses of the moss, *P. patens* PPR (*Pp*PPR) proteins with an evolutionary point of view, relative to the PPR proteins in land plants. *P. patens* (hereafter Physcomitrium) has emerged as a powerful model system in plant functional genomics studies [26,27]. The genome sequences of its plastid, mitochondrion and nucleus have already been disclosed [28,29,30]. Gene targeting is feasible in the nuclear and chloroplast genomes [31,32]. Reverse genetics is usually applied to this moss and stable transformants can be easily generated via homologous recombination [32,33]. Furthermore, the Physcomitrium gametophyte, the haploid phase of the life cycle, is dominant, making it possible to study the phenotypes of the knockouts directly without further crosses. Moreover, the *PPR* gene family is rather compact in Physcomitrium. Therefore, this moss provides researchers with a good tool for the study of PPR, in comparison to flowering plants. To characterize the function of Physcomitrium PPR proteins, my laboratory has constructed gene-targeted knockout or knockdown mutant lines via homologous recombination. To date, 42 *PpPPR* genes were tagged using an antibiotic resistant gene cassette and the characterization of their mutants is in progress. This reverse-genetics approach has thus far revealed the function of 22 *PpPPR* genes.

## 3. The Compact *PPR* Gene Family in Physcomitrium

Moss *PPR* genes were firstly identified in Physcomitrium in 2004 [25]. Thereafter, the whole genome sequence of this moss was disclosed in 2008 [30], and the genome database has been frequently updated. Accordingly, at least 107 *PPR* genes were identified in Physcomitrium, and their encoded proteins were numbered sequentially from PpPPR_1 to PpPPR_107 (Table 1). The *PPR* gene family is rather small in Physcomitrium when it is compared to the large *PPR* gene families in vascular plants [15]. The liverwort, *Marchantia polymorpha,* and the charophytic alga, *Chara braunii,* also possess a compact *PPR* gene family composing between 74 and 57 members, respectively [34,35].

The structure of Physcomitrium *PPR* genes has diverged somewhat from the *PPR* genes in Arabidopsis and rice. Intron-containing *PPR* genes represent three-fourths of all *PPR* genes in Physcomitrium but only one-fourth of them in Arabidopsis and rice [36]. Thus, Physcomitrium PPR genes are generally characterized as intron-rich. The gene structure and encoded amino acid sequence of many PPR proteins are conserved in Physcomitrium and Arabidopsis plants. This conservation suggests that such homologous PPR proteins have the same or similar function in mosses and flowering plants. Presumably, intron-rich *PPR* genes may represent “ancient” *PPR* genes that pre-dated the occurrence of the retrotransposition-mediated expansion of the *PPR* gene family in land plants [36]. Among the 107 PpPPR proteins, at least 18 paralogous pairs have been found (Table 2) and these may have occurred via gene family expansion due to recent genome duplications [30]. Respective paralogous pairs may have redundant or diverse functions.

**Table 1 plants-11-02279-t001:** *Physcomitrium patens* pentatricopeptide repeat (PpPPR) proteins.

Protein Name	Gene Locus ID ^1^	Type ^2^	PPR Motif Bead Patterns and Additional Domain/Motif in Parenthesis ^3^	Subcellular Localization ^4^		Phenotype of Moss Colony of Gene Knockout Lines	Function Identified in *P. patens*	Arabidopsis Homologs	Function Identified in Arabidopsis	Refs.
				Pred	Exp					
PpPPR_1	Pp3c3_19290	P	P-P-P-P-P-P-P-P-P-P-P-P-P	**m**				At5g50280 (EMB1006)	required for embryo development	[14]
PpPPR_2	Pp3c16_9240	P	P-P-P-P-P-P-P-P-P-P-P	c	C	Smaller than WT				
PpPPR_3	Pp3c5_10110	P	(RRM)-P-P-P-P-P-P-P-P-P-P-P-P-P-P-P	other	C	WT-like		At5g04810 (AtPPR4)	*rps12*-intron 1 *trans*-splicing	[37,38]
PpPPR_4	Pp3c17_11510	P	P-P-Pi-P-P-P-P-P-P-P-P	c	C	Very small colony	RNA splicing of group II intron in pre-tRNA^Ile^			[39]
PpPPR_5	Pp3c21_11730	P	P-P-P-P-P-P-P-P-P	m						
PpPPR_6	Pp3c5_21760	P	P-P-P-P	c	C					
PpPPR_7	Pp3c25_10050	P	P-P-P-P-P-P-Pi-P-P-P-(LAGLIDADG)	c				At2g15820 (OTP51)	*ycf3*-intron 2 splicing	[40]
PpPPR_8	Pp3c14_19310	P	P-P-P-Pi-P-P-P-P-P-P-P	m		Smaller than WT				
PpPPR_9	Pp3c24_14870	PLS	P1-L1-S1-P1-L1-S1-P1-L1-S1-P2	m	M	Smaller than WT	RNA splicing of *cox1* intron 3			[41]
PpPPR_10	Pp3c21_550	P	P-P	c				At4g21190 (EMB1417)	chloroplast localized, required for embryo development	[16]
PpPPR_11	Pp3c3_2440	P	P-P-P-P-P	other	M	Smaller than WT	stabilization of *nad7* mRNA			Unpublished
PpPPR_12	Pp3c7_22430	P	P-Pi-P-P-P-P-P-P-P-P-P	c						
PpPPR_13	Pp3c7_17210	P	P-P-P-P-P-P-P-P-P-P-P-P-P-P	c						
PpPPR_14	Pp3c26_10760	P	P-P-P-P-P-P-P-P-P-P-P	c				At3g46610 (LPE1)	binds to the 5′ UTR of *psbA* mRNA	[42]
PpPPR_15	Pp3c17_6450	P	P1-SS-P1-P1-P1	c	C					
PpPPR_16	Pp3c14_20030	P	P-P-P-P-P-P	m						
PpPPR_17	Pp3c8_4580	P	P-P-P-P-P-P-P-P-P-P-P	c	C	Smaller than WT		At4g39620 (AtPPR5)	*trnG* splicing and intron stabilization	[43]
PpPPR_18	Pp3c10_3690	P	P-P-P-P-P-P-P-P-P-P-P-P-P	m						
PpPPR_19	Pp3c10_19570	P	P-Pi-P-P-P-P-P-P-P-P-P-P	c	C	WT-like		At4g34830 (MRL1)	stabilizes *rbcL* 5’ end	[44]
PpPPR_20	Pp3c10_14800	P	P-P-P-P-P-P-P-P	c	C			At1g01970	chloroplast localized, function unknown	[16]
PpPPR_21	Pp3c22_3230	P	P-P-P-P-P-P-P-P-P-P-P-P-P-P-P-P-P-P-P	c	C	Smaller than WT	stablilization of *psbI*-*ycf12* mRNA	At5g02860 (AtPPR21L)	chloroplast localized, probably required for embryo development	[45]
PpPPR_22	Pp3c16_23700	P	P-P-P-P-P-P-P-P-P-P-(LAGLIDADG)	m				At2g15820 (OTP51)	*ycf3* -intron 2 splicing	[40]
PpPPR_23	Pp3c26_11100	P	P-P-P-P-P-P-P-P-P-P	c		Smaller than WT		At3g59040 (PPR287)	crucial for chloroplast function and plant development	[46]
PpPPR_24	Pp3c2_3210	P	P-P-P-P-P-P-P-P-P-P	m						
PpPPR_25	Pp3c16_5160	PLS	P1-L1-S1-P1-L1-S1-P1-L1-S1-P1-L1-S1-P1-L1-S1	m	M					
PpPPR_26	Pp3c10_24680	P	P-P-P-P-P-P-P-P-P	m						
PpPPR_27	Pp3c20_15110	P	P-P-P-P-P-P-P-P-P-P	other	C	WT-like		At3g53170	chloroplast nucleoid	[47]
PpPPR_28	Pp3c6_24160	P	P-P-P-P-P-P-P-P-P-P-P-P	c	C	Smaller than WT				
PpPPR_29	Pp3c5_2770	P	P-P-P-P-P-P-P-P-P-P	m						
PpPPR_30	Pp3c7_10060	P	P-P-P-P-P-P-P-P-(SMR)	m				At1g18900	mitochondrial localized, function unknown	[16]
PpPPR_31	Pp3c6_23550	PLS	L1-S1-P1-L1-S1-P1-L1-S1-P1-L1-S1-P1-L1-S1	m	M	Smaller than WT	RNA splicing of *atp9* intron 1 and *nad5* intron 3			[41]
PpPPR_32	Pp3c8_15500	P	P-P-P-P-P-P-P-P-P-P-P-P-P	c	C	Smaller than WT	*psaC* mRNA accumulation			[48]
PpPPR_33	Pp3c1_37670	P	P-P-P-P-P-P-P-Pi-P-P	m						
PpPPR_34	Pp3c22_1710	PLS	L1-S1-P1-L1-S1-P1-L1-S1-P1-L1-S1-P1-L1-S1-P1-L1-S1-P1-L1-S1	other	C					
PpPPR_35	Pp3c8_13830	P	P-P-P-P-P-P-P-P-P-P	other						
PpPPR_36	Pp3c4_14490	P	P-P-P-P-P	m						
PpPPR_37	Pp3c4_14140	P	P-P-P-P-P-P-P-P-P-P-P-P-P-P-P-P-P-P-P-P-P-P	m	M	Bigger than WT				
PpPPR_38	Pp3c6_26920	P	P-P-P-P-P-P-P-P-P-P-P	c	C	Very small colony	RNA processing of *clpP*-5′-*rps12* mRNA			[49,50]
PpPPR_39	Pp3c4_3090	P	P-P-P-P-P-Pi-P-P	c				At3g42630 (PBF2)	RNA splicing of *ycf3*	[51]
PpPPR_40	Pp3c23_13490	P	P-P-P-P-P-P-P-P-P	c	C			At5g67570 (DG1)	involved in the regulation of early chloroplast development	[52]
PpPPR_41	Pp3c13_18720	P	P-Pi-P-P-P-P	other	C					
PpPPR_42	Pp3c21_12360	P	P-P-P-P-P-P-P-P-P-P-(SMR)	m				At5g02830	chloroplast nucleoid	[47]
PpPPR_43	Pp3c3_24770	DYW	L1-S1-P1-L1-S1-P1-L1-S1-P1-L1-S1-P1-L1-S1-P1-L1-S1-P1-L1-S1-P2-L2-S2-E1-E2-DYW	m	M	Very small colony	RNA splicing of *cox1* intron 3			[53]
PpPPR_44	Pp3c17_24090	P	P-P-P-P-P-P-Pi-P-P-P	m						
PpPPR_45	Pp3c11_7720	DYW	L1-S1-P1-L1-S1-P1-L1-S1-P1-L1-S1-P1-L1-S1-P1-L1-S1-P1-L1-S1-P2-L2-S2-E1-E2-DYW	c	C		RNA editing at *rps14*-C2			[54]
PpPPR_46	Pp3c3_5760	P	P-P-P-P-P-P-P-P-P-P-P-P-P-P-P	c				At5g39980 (PDM3)	essential for chloroplast development	[55]
PpPPR_47	Pp3c4_17980	P	P-Pi-P-P-P-P-P-P-P-P	other						
PpPPR_48	Pp3c24_4430	P	P-P-P-P-P-P-P-P-P-P-P-P-Pi-P-P	c	M	WT-like		At1g10910 (PDM2)	involved in the regulation of chloroplast development	[56]
PpPPR_49	Pp3c1_6490	P	P-P-P-P-P-P-P-P-P-Pi-P-P-P-P-P-P-P-P-P-P-P-P-P-P	m	M	WT-like				
PpPPR_50	Pp3c11_2930	P	P-P-P-Pi-P-Pi-Pi-P-P-P-P-P-P-P-P-P	m						
PpPPR_51	Pp3c12_14320	P	P-P-P-P-P-P-P-P-P-P	c	C			At4g34830 (MRL1)	stabilizes *rbcL* 5′ end	[44]
PpPPR_52	Pp3c25_7340	P	P-P-P-P-P	c	C					
PpPPR_53	Pp3c13_17120	P	P-P-P-P-P-P-P-P-P-P	c	C	WT-like		At5g02860	chloroplast nucleoid	[47]
PpPPR_54	Pp3c12_26140	P	P-P	m	C/M	Bigger than WT				
PpPPR_55	Pp3c6_14920	P	P-P-P	c				At3g46870 (THA8-like)	*ycf3*-intron 2 and *trnA* splicing	[57]
PpPPR_56	Pp3c19_930	DYW	L1-S1-P1-L1-S1-P1-L1-S1-P1-L1-S1-P2-L2-S2-E1-E2-DYW	m	M	WT-like	RNA editing at *nad3*-C230, *nad4*-C272			[58]
PpPPR_57	Pp3c12_4690	P	P-P-P-P-P-P-P-P-P	m						
PpPPR_58	Pp3c1_21850	P	P-P-P-P-Pi-Pi-P-P-P-P	m				At4g35850	present in mitochondrial complexome	[59]
PpPPR_59	Pp3c22_3070	P	P-P-P-P-P-P-P-P-P-P-P-P-P (SMR)	c	C	WT-like		At5g02830	chloroplast nucleoid	[47]
PpPPR_60	Pp3c16_9420	P	P-P-P-P-P-P-P-P-P-P	m						
PpPPR_61	Pp3c14_7210	P	P-P-P-P-Pi-Pi-P-P-P-P	m				At4g35850	present in mitochondrial complexome	[59]
PpPPR_62	Pp3c16_9880	P	P-P-P-P-(SMR)	m				At2g17033	chloroplast localized, function unknown	[16]
PpPPR_63	Pp3c7_17100	P	P-Pi-P-(NYN)	other	NUC	Very small colony	5’-end processing of pre-tRNA	At2g16650 (PRORP2)At4g21900 (PRORP3)	5’-end processing of pre-tRNA	[60,61]
PpPPR_64	Pp3c11_11830	P	P-P-P-P-P-P-P-P-P-P-P-P-P-P-P-Pi-P-(SMR)	other	C	Smaller than WT	Expression of *psaA*-*psaB*-*rps14*	At1g74850 (pTAC2)	involved in transcription by PEP	[62,63]
PpPPR_65	Pp3c4_16600	DYW	S2-L1-S1-P1-L1-S1-P1-L1-S1-P1-L1-S1-P2-L2-S2-E1-E2-DYW	m	M	Very small colony	RNA editing at *ccmFc*-C103			[64,65]
PpPPR_66	Pp3c16_5890	P	P-P-P-P-P-P-P-P-P-P-P	c	C	WT-like	RNA splicing of *ndhA*	At2g35130 (AtPPR66L)	RNA splicing of *ndhA*	[66]
PpPPR_67	Pp3c2_30390	P	P-P-P-(NYN)	other	C/M	WT-like	5′-end processing of pre-tRNA	At2g32230 (RPORP1)	5′-end processing of pre-tRNA	[60,61]
PpPPR_68	Pp3c2_27580	P	P-P-P-P-P-P-P-P-P-P-P-P-P-P	m						
PpPPR_69	Pp3c17_5040	PLS	L1-S1-P1-L1-S1-P1-L1-S1-P1-L1-S1-P1-L1-S1-P2	sp	C	WT-like		At4g18520 (PDM1/SEL1)	RNA editing of *accD*, *trnK*-intron splicing, RNA processing of *rpoA*	[67]
PpPPR_70	Pp3c8_4280	P	P-P-P-P-P-P-P-P-P-(CBS)	c	C/NUC			At5g10690 (CBSPPR1)	chloroplast nucleoid, possibly play a role in regulating transcription or replication	[47,68]
PpPPR_71	Pp3c14_16110	DYW	L1-S1-P1-L1-S1-P1-L1-S1-P1-L1-S1-P1-L1-S1-P2-L2-S2-E1-E2-DYW	m	M	Very small colony	RNA editing at *ccmFc*-C122			[69]
PpPPR_72	Pp3c6_26210	P	P-P-P-P-P-P-P-P-P-P-P	c	C/M		function unknown	At2g35130 (AtPPR66L)	RNA splicing of *ndhA*	[66]
PpPPR_73	Pp3c3_31860	P	P-P-P-P-P-Pi-Pi-P	m	C/M					
PpPPR_74	Pp3c17_13520	P	P-P-P-P-P-P-P-P-P-P-P-P-P-P-P-P-P-P-P-P-P-P-P-P	c	C	Smaller than WT				
PpPPR_75	Pp3c5_16950	P	P-P-P-P-P-P-P-P-P-P-P-P-P-P-(SMR)	c	C			At2g31400 (GUN1)	involved in retrograde signaling to the nucleus.	[70]
PpPPR_76	Pp3c16_280	P	(RRM)-P-P-P-P-P-P-P-P-P-P-P-P-P-P-P	other	C			At5g04810 (AtPPR4)	*rps12*-intron 1 *trans*-splicing	[37,38]
PpPPR_77	Pp3c5_15090	DYW	L1-S1-P1-L1-S1-P1-L1-S1-P1-L1-S1-P1-L1-S1-P1-L1-S1-P1-L1-S1-P1-L1-S1-P2-L2-S2-E1-E2-DYW	other	M	Very small colony	RNA editing at *cox2*-C370, *cox3*-C733			[71]
PpPPR_78	Pp3c2_12230	DYW	L1-S1-P1-L1-S1-P1-L1-S1-P1-L1-S1-P1-L1-S1-P1-L1-S1-P2-L2-S2-E1-E2-DYW	other	M	WT-like	RNA editing at *rps14*-C137, *cox1*-C755			[71,72]
PpPPR_79	Pp3c5_7610	DYW	L1-S1-P1-L1-S1-P1-L1-S1-P1-L1-S1-P1-L1-S1-P2-L2-S2-E1-E2-DYW	m	M	Smaller than WT	RNA editing at *nad5*-C598			[58]
PpPPR_80	Pp3c4_9690	P	P-P-P-P-P-P-P-P-P-P	c	C			At4g39620 (AtPPR5)	*trnG* splicing and intron stabilization	[43]
PpPPR_81	Pp3c14_15490	P	P-P-P-P-P-P-P-P-P-P-P-P-P-P-P-(SMR)	other	C	Bigger than WT				
PpPPR_82	Pp3c9_6880	P	P-P-P-P-P-P-P-P-P	other	C					
PpPPR_83	Pp3c2_1940	P	P-P-P-P-P-P-P-P-P-P-P-P-P-P-P-Pi-P-P-P	other				At2g41720 (EMB2654)	*rps12*-intron 1 trans-splicing	[73,74]
PpPPR_84	Pp3c15_12010	P	P-P-P-P-P-P-P-P-P-P	c						
PpPPR_85	Pp3c6_15900	P	P-P-P-P-P-P-P-P-P-P-P-P-P-P-(SMR)	c	C	WT-like		At2g31400 (GUN1)	involved in retrograde signaling to the nucleus	[70]
PpPPR_86	Pp3c17_2130	P	P-P-Pi-P-Pi-P-P-P-P-P-P-P-P-P-P-P-P-Pi-P	c						
PpPPR_87	Pp3c8_15040	P	P-P-P-P-P-P-P-P-P-P-P-P-P	m						
PpPPR_88	Pp3c18_8600	P	P-P-P-P-P-P-P-P-P	m						
PpPPR_89	Pp3c1_28760	P	P-P-P-P-P-P-P	m	M					
PpPPR_90	Pp3c11_21930	15P	P-P-Pi-P-P-P-P-P-P-P-P-P-P-P-P	other				At5g42310 (AtCRP1)	stabilizes 5′ and 3′ ends in *petB*-*petD* intergenic region; activates *petA*, *psaC*, and *petD* translation	[17,18,75,76]
PpPPR_91	Pp3c17_23250	DYW	P1-L1-S1-P1-L1-S1-P1-L1-S1-P2-P1-L1-S1-S1-P1-L1-S1-P2-L2-S2-E1-E2-DYW	m	M	Very small colony	RNA editing at *nad5*-C730			[58]
PpPPR_92	Pp3c5_2530	P	P-P-P-P-P-P-P-Pi-P-P-P-P-P-P-P-P-P-P-P-P	c	C	Smaller than WT		At4g30825 (BFA2)	accumulation of the *atpH/F* transcript	[77]
PpPPR_93	Pp3c6_3910	P	P-P-P-P-P-P-P-P-P-P-P-P-P-P	other						
PpPPR_94	Pp3c16_4140	P	P-P-P-P-P-P-P-P-P-P-P-P-P-P-P-P-P-P-P-P-P-P	c	C			At4g30825 (BFA2)	accumulation of the *atpH*/*F* transcript	[77]
PpPPR_95	Pp3c5_26320	P	P-P-P-P-P-P-P-P-P-P-P-P-P-P	c						
PpPPR_96	Pp3c4_4900	P	P-P-P-P-P-P-P-P-P-P-Pi-P-(SMR)	c	C					
PpPPR_97	Pp3c3_19780	P	P-P-P-P-P-P-P-P-P-P-P-P-P-P	m	M					
PpPPR_98	Pp3c27_5540	DYW	P1-L1-S1-P1-L1-S1-P1-L1-S1-P1-L1-S1-P1-L1-S1-P1-L1-S1-P2-L2-S2-E1-E2-DYW	m	M	WT-like	RNA editing at *atp9*-C92			[64,65]
PpPPR_99	Pp3c12_2390	P	P-P-P-P-P-P-P-P-P-P-P-P-P	c	C			At3g09650 (HCF152)	stabilizes 5′ and 3′ ends in *psbH*-*petB* intergennic region, also stimulates *petB* splicing	[20,78]
PpPPR_100	Pp3c12_8330	P	P-P-P-P-Pi-Pi-P-P-P	c				At2g30100	chloroplast localized, function unknown	[16]
PpPPR_101	Pp3c2_36070	P	P1-P1-P2-L2-P1-P1-P2-P1-P1-P1-P1-L2-P1-P1	m						
PpPPR_102	Pp3c6_11500	P	P-P-P-P	c	C	WT-like				
PpPPR_103	Pp3c17_4890	P	P1-L1-P2	other						
PpPPR_104	Pp3c10_16850	P	P-P-Pi-P-(NYN)	m	C/M	Smaller than WT	5’-end processing of pre-tRNA	At2g32230 (RPORP1)	5’-end processing of pre-tRNA	[60,61]
PpPPR_105	Pp3c24_8560	PLS	L1-S1-P1-L1-S1-P1-L1-S1-P1-L1-S1-P1-L1-S1-P2-S1-P1-L1-S2	m	C	WT-like				
PpPPR_106	Pp3c22_3080	P	P-P-(SMR)	sp						
PpPPR_107	Pp3c1_8170	P	P-P-P-P-P-P-(SAP)-P	m	C			At3g04260 (pTAC3)	light-dependent transcription	[79]

^1^ Gene locus ID is from *P. patens* genome release v3.3 (https://phytozome-next.jgi.doe.gov (accessed on 1 July 2022)). ^2^ PPR types are according to Lurin et al. [14] and Cheng et al. [15]: “P” for the P type, “PLS” for the PLS type containing P1, P2, L1, L2, S1, S2 and “DYW” for the DYW type containing E1, E2 and DYW domain. ^3^ Motif bead patterns are according to Cheng et al. [15] (https://ppr.plantenergy.uwa.edu.au (accessed on 1 July 2022)). Additional domains or motifs are RRM (RNA recognition motif), LAGLIDADG (LAGLIDADG RNA motif), SMR (small Mut-related), NYN (Nedd4-BP1, YacP nucleases domain), CBS (cystathionine β-synthase domain) and SAP (SAF-A/B, Acinus and PIAS domain). ^4^ Subcellular localization. Predicted localizations (Pred) were provided by the TargetP server v1.1 (http://www.cbs.dtu.dk/services/TargetP-1.1/index.php (accessed on 1 July 2022)). The prediction is noted in lowercases as follows: m, mitochondria; c, chloroplasts; sp, signal peptide; other, any other location. Experimental localizations (Exp) of fluorescent proteins were from published studies and unpublished studies of our laboratory. The conclusion is indicated in uppercases as follows: M, mitochondria; C, chloroplasts; C/M, chloroplasts and mitochondria; NUC, nucleus; C/NUC, chloroplasts and nucleus.

## 4. Subcellular Localization of PPR Proteins in Physcomitrium

In silico and in vivo analyses have shown that 97 out of 107 PpPPR proteins are presumably localized in either the plastids or mitochondria, or both (Table 1). An in vivo analysis using a transient assay or transgenic Physcomitrium plants expressing the PpPPR-green fluorescent protein (GFP) fusion protein demonstrated the subcellular localization of 60 PpPPR proteins, wherein 38 of which are chloroplast-targeted, 17 are localized in mitochondria, and four are targeted to both. PpPPR_63 is localized in the nucleus and its paralogs (PpPPR_67 and 104) are localized in both the chloroplasts and mitochondria [60]. Thirteen PpPPR proteins are predicted to be localized in the plastids and 24 in the mitochondria, while the location of the remaining 10 has not yet been predicted.

## 5. P-Type of PPR Proteins in Physcomitrium

More than half (55%) of the Arabidopsis *PPR* genes encode P-type proteins while most (85%) of the Physcomitrium PPR proteins are of the P type. About 40% of Physcomitrium P-type proteins show high aa identities (30–50%) with Arabidopsis PPR proteins, such as EMBRYO DEFECTIVE (EMB)1006 [14], AtPPR4 [37,38], AtPPR5 [43], Maturation/stability of *RbcL* 1 (MRL1) [44], THYLAKOID ASSEMBLY 8 (THA8)-like [57], GENOME UNCOUPLED 1 (GUN1) [70], plastid Transcriptionally Active Chromosome 2 (pTAC2) [62], ORGANELLE TRANSCRIPT PROCESSING 51 (OTP51) [40], and Proteinaceous RNase P (PRORP) 1, 2 and 3 [61] and others [42,46,47,51,52,55,56,59,67,68,73,74,75,76,77,79] (Table 1). The remaining 60% of the P-type PpPPR proteins may be unique to this moss but their functions are totally unknown.

The P-type PPR proteins are mostly involved in intergenic RNA processing, RNA splicing, RNA stabilization, as well as translation initiation [8,9,10,43,80]. In all these cases, PPR proteins bind in a gene-specific manner to target RNAs. Bioinformatic analyses of PPR proteins and target RNA sequences have proposed an RNA recognition code for PPR proteins involving a combination of amino acid residues at two or three positions in a PPR motif that determines the base preference [81,82,83]. The crystal structures of THA8 and PPR10 in an RNA-bound state were determined [84,85]. The molecular basis for the specific and modular recognition of RNA bases, A, G and U was revealed. The structural elucidation of RNA recognition by PPR proteins provides an important framework for the potential biotechnological applications of PPR proteins in RNA-related research areas [86,87,88].

### 5.1. P-Type Proteins Involved in RNA Processing/Stabilization

#### 5.1.1. PpPPR_38

In Physcomitrium, the first functional analysis of a PPR protein was achieved for PpPPR_38 which has 10 P motifs [49]. PpPPR_38 is localized in the chloroplasts and is responsible for the intergenic RNA cleavage between *clpP* and 5′-*rps12,* encoding the ATP-dependent protease proteolytic subunit and the ribosomal protein S12, respectively. The *clpP* gene, which contains two introns, is co-transcribed with the 5′-*rps12* and *rpl20* genes, and a primary transcript of 3.2 kb is produced. The primary transcript is then cleaved at the intergenic spacer between *clpP* and 5′-*rps12* and is spliced to produce a 0.6-kb mature *clpP* mRNA. This post-transcriptional event proceeds rapidly and therefore the primary transcript accumulates in small amounts. In the *PpPPR_38* knockout (KO) mutants, the primary transcript accumulates at a substantial level while mature *clpP* mRNA is significantly decreased [49]. PpPPR_38 binds specifically to the intergenic region of *clpP*–5′-*rps12* dicistronic mRNA and may stabilize a processed *clpP* 3′-end [50]. The binding of the PpPPR_38 protein could recruit endonucleases and thus may lead to RNA cleavage. Interestingly, the processed *rps12* 5′-ends in moss and Arabidopsis were mapped at analogous positions and the sequence corresponding to the overlap between the processed *clpP* and *rps12* mRNAs shows a striking conservation between angiosperms and moss [78]. These findings suggest that an unidentified conserved protein, possibly a PpPPR_38 ortholog, binds to the *clpP–rps12* intergenic region and blocks the RNA degradation in angiosperms.

#### 5.1.2. PpPPR_21

PpPPR_21 is comprised of 19 P motifs and is chloroplast-localized. The KO mutants of *PpPPR_21* grow slowly and exhibit a significant reduction of the photosystem II (PSII) core protein D1 (PsbA) level, and a concomitantly poor level of the PSII supercomplexes [45]. Similar phenotypic features were reported in the green alga, *Chlamydomonas reinhardtii* (Chlamydomonas) and tobacco, *psbI* gene KO mutants [89,90]. Tobacco, *psbI,* KO mutants are autotrophically viable, but PsbA and PsbO levels are reduced to 50% of the level of the wild type (WT) while the PSII complexes are poorly formed, suggesting that PsbI is essential for the stability of dimeric PSII and supercomplexes [45]. Notably, the *psbI*-*ycf12* dicistronic mRNA (encoding PSII I-subunit and YCF12) is lost in the *PpPPR_21* KO mutants while other PSII gene transcript levels are not altered in these mutants [45]. An RNA electrophoresis mobility shift assay (REMSA) showed that the recombinant PpPPR_21 binds efficiently to the 5′ untranslated region (UTR) of *psbI* mRNA [45]. These observations suggest that PpPPR_21 is involved in the accumulation of a *psbI-ycf12* mRNA.

PpPPR_21 homologs, which are referred to as PPR21L, are widely distributed in bryophytes, ferns and seed plants, but not in Chlamydomonas. PpPPR_21 shows 45% aa identity and 85% similarity to Arabidopsis At5g02860 (AtPPR21L), which is predicted to be localized in the chloroplasts [16]. A predicted target sequence of Arabidopsis At5g02860 is found in the 5′-UTR of the Arabidopsis *psbI* transcript. Unfortunately, no homozygous mutants of T-DNA-tagged *At5g02860* lines have been obtained, so the loss of function of *AtPPR21L* may lead to embryonic lethality. To confirm whether AtPPR21L is a functional ortholog of PpPPR_21, other *AtPPR21L* mutants will be generated and characterized, for example, by generating knockdown mutants.

#### 5.1.3. PpPPR_32

PpPPR_32 consists of 13 P motifs and is localized in the chloroplasts [48]. *PpPPR_32* homologs are not found in seed plants and are unique to early land plants. *PpPPR_32* KO mutants grow autotrophically but they have reduced protonema growth and the poor formation of photosystem I (PSI) complexes. In addition, a significant reduction in the transcript level of the *psaC* gene encoding the iron sulfur protein of PSI was observed in the KO mutants, but the transcript levels of other PSI genes were not altered. This indicates that PpPPR_32 is essential for the accumulation of *psaC* transcript and PSI complexes. In land plants, *psaC* is organized as a gene cluster with *ndhH, A, I, G, E* and *D* genes and is located between *ndhE* and *D*. These genes are co-transcribed as a long primary transcript (>7 kb) and then they are post-transcriptionally processed to multiple and overlapping intermediates and mature transcripts in seed plants [91,92]. The monocistronic *psaC* transcript is produced from a *psaC*-*ndhD* dicistronic mRNA by the cleavage of its intergenic region, and then it represents the translatable mRNA [91,93]. Based on the stoichiometry of accumulating PSI and NDH complexes [94], the expression of *psaC* is two orders higher than that of *ndhD* in higher plants. Likewise, the monocistronic *psaC* mRNA is extremely stable, more so than *ndh* transcripts in Physcomitrium chloroplasts.

PsaC is an essential component, both for photochemical activity and for the stable assembly of PSI in Chlamydomonas [95] and higher plants [96]. The loss of PsaC leads to defects in PSI activity and the rapid degradation of other subunits within the core complex. A reduction in the *psaC* transcript level may cause a reduction in the PSI subunits in *PpPPR_32* KO mutants. An in silico analysis predicted that PpPPR_32 binds to one of two stem-loop structures in the 3′-UTR of *psaC* mRNA [48]. The binding of PpPPR_32 to the stem-loop may facilitate the stabilization of *psaC* mRNA. The binding of PpPPR_32 could prevent endonucleolytic cleavage by blocking the access of RNases, thus stabilizing the RNA. This binding might occur in the vicinity of secondary structure elements, such as hairpins, that are known targets for endonucleases.

### 5.2. P-Type Proteins Involved in RNA Splicing

Several P-type PPR proteins that are involved in the splicing of plastid group II introns have been identified. Maize PPR4, with an N-terminal RNA recognition motif (RRM), facilitates *rps12 trans*-splicing through direct interaction with intron RNA [37]. Maize PPR5 is involved in the splicing or the stability of pre-tRNA^Gly^ [43]. Arabidopsis OTP51 with two C-terminal LAGLIDADG motifs is required for the *cis*-splicing of the *ycf3*-2 intron [40]. Maize and Arabidopsis THA8 is essential for the splicing of both *ycf3*-2 and tRNA^Ala^ introns [57]. PPR4, PPR5, OTP51 and THA8 homologs were found in Physcomitrium (Table 1) but their function has yet to be elucidated. To date, PpPPR_4 and PpPPR_66 were identified as P-type proteins that are required for RNA splicing in plastid transcripts [39,66].

#### 5.2.1. PpPPR_4

The ten P motif-containing PpPPR_4, which is not related to maize PPR4, is required for the group II intron splicing of pre-tRNA^Ile^-GAU [39]. The tRNA^Ile^ gene is co-transcribed as a long primary transcript from a *rrn16*-*trnI(GAU)*-*trnA(UGC)*-*rrn23*-*rrn4.5* gene cluster, then an intron-containing pre-tRNA^Ile^ is produced and spliced to produce a mature tRNA^Ile^. *PpPPR_4* KO mutants display severe growth retardation, a reduced effective quantum yield of PSII and a strongly reduced level of plastid-encoded proteins, such as PSII D1 (PsbA) protein, the β subunit of ATP synthase, and the stromal enzyme, Rubisco. Analyses of the chloroplast transcriptome revealed that the disruption of *PpPPR_4* resulted in a significant reduction of the mature tRNA^Ile^ (GAU) and the aberrant accumulation of intron-containing pre-tRNA^Ile^. The recombinant PpPPR_4 was shown to bind preferentially to domain III of the tRNA^Ile^ group II intron [39]. Thus, the *PpPPR_4* gene KO specifically impaired the splicing of tRNA^Ile^ (GAU), but it did not affect tRNA^Ala^ (UGC) splicing. This is distinct from maize and Arabidopsis *tha8* mutants, where tRNA^Ala^ splicing was strongly inhibited, whereas tRNA^Ile^ splicing was not affected [57].

In angiosperms, the splicing of pre-tRNA^Ile^ is known to require at least four nuclear-encoded factors, ribonuclease III domain-containing protein RNC1 [97], WTF1 (a protein harboring a “domain of unknown function 860”, assigned the name WTF1 “what’s this factor?”) [98], DEAD box RNA helicase RH3 [99], mitochondrial transcription termination factor 4 (mTERF4) [100] and the chloroplast *trnK* intron-encoded maturase (MatK) [101]. These are thought to be major splicing factors and constitute a spliceosome-like complex that is for splicing chloroplast group II introns [102]. The Physcomitrium genome encodes an RNC1 homolog, a WTF1 homolog, four RH3 homologs and three mTERF4 homologs. It will be important in the future to examine whether these homologs are involved in the splicing of tRNA^Ile^ (GAU) in Physcomitrium. Intriguingly, a PpPPR_4 homolog was found in a lycophyte, *Selaginella moellendorffii*, but not in Arabidopsis and rice. Presumably, the Arabidopsis PPR protein(s) involved in the splicing of pre-tRNA^Ile^ (GAU) may have lost their importance during evolution, whereas other proteins may have replaced PpPPR_4.

#### 5.2.2. PpPPR_66

PpPPR_66 is a chloroplast-localized protein with 11 P motifs. *PpPPR_66* KO mutants exhibited a WT-like growth phenotype, but they showed aberrant chlorophyll fluorescence due to defects in the chloroplast NADH dehydrogenase-like (NDH) activity [66]. In addition, the chloroplast NDH complex was completely lost in the KO mutants. Among the 11 *ndh* genes in the chloroplast genome, *ndhA* expression was specifically affected in the KO mutants. The chloroplast *ndhA* gene contains a group II intron and its intron splicing was almost completely blocked in the *PpPPR_66* KO mutants. PpPPR_66 binds to the 5′ half of domain I of the *ndhA* group II intron [66].

PpPPR_66 shows 44% aa identity and 81% similarity to the Arabidopsis PPR protein (At2g35130). The *At2g35130* gene is interrupted by seven introns and their positions are identical to the intron positions of the *PpPPR_66* gene, suggesting that At2g35130 is an ortholog of PpPPR_66. In fact, among the Arabidopsis *At2g35130* KO null mutant lines, SALK_043507 and SALK_065137 exhibited defects in chloroplast NDH activity and *ndhA* splicing [66]. PpPPR_66-like homologs (PPR66L) are widely distributed in land plants but not in green algae, *Chlamydomonas*, *Volvox* or *Chlorella*. Streptophyta (charophytes and land plants) have *ndhA* genes with a group II intron, while green algae have an intron-less *ndhA* gene in the chloroplast genome. Thus, there is likely coevolution of PPR66L and the *ndhA* intron in the land plant lineage. Although Arabidopsis At2g35130 (AtPPR66L) was involved in *ndhA* splicing, *AtPPR66L* cDNA did not rescue the *ndhA* splicing deficiency in the *PpPPR_66* KO mutant [66]. Thus, PpPPR_66 and AtPPR66L are required for *ndhA* splicing but their mode of action for splicing the *ndhA* transcript might differ slightly between Physcomitrium and Arabidopsis. This possibility still needs to be assessed.

In seed plants, the splicing of the *ndhA* transcript is known to require several nuclear-encoded factors, including CHLOROPLAST RNA SPLICING 2 (CRS2) [103], CRS2-ASSOCIATED FACTOR 1 and 2 (CAF1 and CAF2) [104] and CHLOROPLAST RNA SPLICING AND RIBOSOME MATURATION (CRM) FAMILY MEMBER 2 (CFM2) [105]. These splicing-related factors have not been identified in Physcomitrium.

### 5.3. P-Type Proteins with an NYN Domain

RNase P is a ubiquitous endonuclease that removes the 5′ leader sequence from pre-tRNAs in all organisms [106]. In Arabidopsis, RNA-free proteinaceous RNase P1 (PRORP1), PRORP2 and PRORP3 were shown to be enzyme(s) for pre-tRNA 5′-end processing in organelles and the nucleus [61,107]. PRORPs contain three P motifs and an NYN (Nedd4-binding protein 1 and the bacterial YacP-like metallonuclease nucleases) domain [108]. PRORP1 is localized in both the chloroplasts and the mitochondria, whereas PRORP2 and 3 are localized in the nucleus. In Physcomitrium, three PRORP-like proteins, PpPPR_63, 67 and 104, were identified [60]. PpPPR_63 was localized in the nucleus, while PpPPR_67 and 104 were found in both the chloroplasts and the mitochondria. The three proteins exhibited pre-tRNA 5′-end processing activity in vitro [60]. *PpPPR_63* KO mutants display the growth retardation of protonema colonies, indicating that unlike Arabidopsis nuclear RPORP2 and 3, the moss nuclear PpPPR_63 is not essential for viability, but it is involved in the growth and development of protonema filaments. In the KO mutants, the level of nuclear-encoded tRNA^Asp^ (GUC) decreased slightly, whereas the levels of most nuclear-encoded tRNAs were not altered. This indicates that most of the cytosolic mature tRNAs are produced normally, without proteinaceous RNase P-like PpPPR_63. Single *PpPPR_67* or *104* gene KO mutants display different phenotypes of protonema growth and plastid tRNA^Arg^ (ACG) accumulation. However, the levels of all other tRNAs are not altered in the KO mutants. In addition, *in vitro* RNase P assays showed that PpPPR_67 and PpPPR_104 efficiently cleaved plastid pre-tRNA^Arg^ (CCG) and pre-tRNA^Arg^ (UCU), but they cleaved pre-tRNA^Arg^ (ACG) with different efficiency [60]. This suggests that the two proteins have an overlapping function, whereas their substrate specificity is not identical.

### 5.4. P-Type Proteins with an SMR Domain

DNA-damage repair systems, including the mismatch-repair pathway, operate in prokaryotes and eukaryotes. The mismatch-recognizing protein, MutS, is central to the mismatch-repair pathway because it recognizes and binds to mismatched nucleotides. The *mutS* and *mutS2* genes exist in a variety of bacterial species and their eukaryotic homologues form a multigene family. Novel, small proteins containing the MutS2 C-terminus were also found in bacteria and fungi and they were named as small MutS-related (SMR) proteins [109]. SMR domain-containing PPR proteins, referred as PPR-SMR proteins, constitute a small family in higher plants, with eight PPR-SMR proteins in Arabidopsis and maize [110]. Arabidopsis PPR-SMR protein GUN1 is involved in plastid-to-nucleus retrograde signaling [70], and SUPPRESSOR OF VARIEGATION7 (SVR7) could be required for FtsH-mediated chloroplast biogenesis [111]. Like SVR7, Arabidopsis SUPPRESSOR OF THYLAKOID FORMATION 1 (SOT1) facilitates the correct processing of plastid 23S–4.5S rRNA precursors [112]. Maize PPR53, an ortholog of Arabidopsis SOT1, plays a role in the expression of the *ndhA* and 5′ end processing of the 23S rRNA precursor [113]. The SMR domain of SOT1 conferred RNA endonucleolytic activity [114]. Thus, PPR-SMR proteins play essential roles in embryo development, chloroplast biogenesis and gene expression [110,115].

In Physcomitrium, 10 PPR-SMR proteins were identified, of which PpPPR_59, 64, 75, 81, 85 and 96 targeted the chloroplasts. PpPPR_75 and 85 were previously assigned as a GUN1 homolog, and PpPPR_64 was assigned as a pTAC2 homolog [116] (Table 1). The loss of *pTAC2* (*ptac2*) results in pale yellow-green primary leaves and a seedling-lethal phenotype in Arabidopsis [62]. Plastid-encoded RNA polymerase (PEP) and its accessary polypeptides are either absent or strongly reduced in the *ptac2* mutant. Similar phenotypes were shown in the rice mutant, *OspTAC2* [117]. In contrast, *PpPPR_64* KO mutants grow autotrophically but more slowly than the WT do [63]. Even though the levels of PSI and PSII are considerably reduced in the KO mosses, unlike in Arabidopsis *ptac2*, most PEP-dependent plastid transcripts, including *psbA*, accumulate at similar levels in the WT and KO mosses. However, the transcript level of the *psaA-psaB-rps14* gene cluster was significantly reduced in the KO mutants. Thus, PpPPR_64 is not likely a functional orthologue of pTAC2.

Single KO mutants of *PpPPR_59* and *85* exhibited no obvious external phenotypic differences in their protonema growth when compared to that of WT moss plants. PpPPR_75 is highly conserved (91.5% aa identity) with PpPPR_85 (Table 2), so it may complement with the loss-of-function of PpPPR_85. Interestingly, KO mutants of *PpPPR_81* have a larger protonema colony than WT do.

## 6. PLS-Type PPR Proteins in Physcomitrium

PLS-type PPR proteins represent a small group in the PLS-subfamily in Arabidopsis and rice [15]. The function of most PLS-type proteins remains elusive. Arabidopsis chloroplast-localized PLS-type protein Pigment-Deficient Mutant 1 (PDM1)/SEEDLING LETHAL1 (SEL1) is reported to influence the RNA editing of the *accD* transcript encoding the acetyl-CoA carboxylase β subunit [67,118] and the splicing of *trnK* transcript encoding tRNA^Lys^ and *ndhA* transcripts [67]. Rice PLS-type protein PALE GREEN LEAF12 (PGL12) is also a chloroplast-localized protein that was involved in the processing of 16S rRNA and the splicing of the *ndhA* transcript [119].

The Physcomitrium nuclear genome encodes six PLS-type proteins, PpPPR_9, 25, 31, 34, 69 and 105 (Table 1). PpPPR_34, 69 and 105 are targeted to chloroplasts while the others are mitochondria-localized. PpPPR_69 seems to be a PDM1/SEL1 homolog, but the remaining five proteins are structurally unrelated to the Arabidopsis and rice PLS-type proteins. KO mutants of *PpPPR_69* or *105* exhibit no obvious external phenotypic differences when they are compared to WT. In contrast, the KO mutants of *PpPPR_9* and *31* encoding the mitochondrial localized PLS-type protein display slower protonema growth than the WT does [41]. The *PpPPR_31* KO mutants show a considerable reduction in the splicing of *nad5* intron 3 and *atp9* intron 1 (Figure 3). The *PpPPR_9* KO mutants display a severe reduction in the splicing of *cox1* intron 3 (Figure 3). Their intron splicing efficiency in the KO mutants was reduced to less than 50% relative to that in the WT. In Physcomitrium, the splicing of *cox1* intron 3 required DYW-type PpPPR_43 [53]. The splicing of *cox1* intron 3 was completely abolished in the *PpPPR_43* KO mutants while it occurred—but at a lower efficiency—in the *PpPPR_9* KO mutants. The loss of *PpPPR_43* resulted in severe growth retardation while the knockout of *PpPPR_9* only led to the generation of smaller protonema colonies when it is compared to WT. This phenotypic difference can be explained by the different splicing efficiencies of *cox1* intron 3 in both mutants. PpPPR_43 may contribute as a major factor to the splicing of *cox1* intron 3, and PpPPR_9 may assist its splicing as an auxiliary factor.

In Arabidopsis mitochondria, nine genes are interrupted by 23 group II introns that are removed out, as 18 undergo *cis*-splicing and five undergo *trans*-splicing. In Physcomitrium, there are 16 intron-containing genes that are interrupted by 25 group II and two group I introns [29]. All of these introns are spliced out by *cis*-splicing. The mitochondrial genes, *cox1* and *atp9,* have four and three introns in Physcomitrium but no introns in Arabidopsis. The mitochondrial *nad5* gene is interrupted by three introns in Physcomitrium. The first and third introns are group II introns, and the second intron is a group I intron. In contrast, the Arabidopsis *nad5* gene is interrupted by two *cis*- and two *trans*-group II introns. Thus, the positions of the intron insertion sites in the mitochondrial genes are completely different between Physcomitrium and Arabidopsis. Hence, PpPPR_9, 31 and 43 are Physcomitrium-specific PPR splicing factors. PLS-type PPR proteins that are involved in the splicing of other introns of either plastids or mitochondrial genes have yet to be identified.

## 7. DYW-Type PPR Proteins in Physcomitrium

Physcomitrium has 10 DYW-type PPR proteins, PpPPR_43, 45, 56, 65, 71, 77, 78, 79, 91 and 98 (Table 1). In Arabidopsis, 82 DYW domain-containing proteins were identified and many of them are involved in RNA editing [15]. The DYW domain was named after its characteristic C-terminal tripeptide, Asp (D)-Tyr (Y)-Trp (W) and is not found in any other proteins or organisms apart from land plants, except for a heterolobosean protist, *Naegleria gruberi* [120]. The protist DYW-type proteins are hypothesized to derive from horizontal gene transfer from plants in very early land plant evolution. DYW-type proteins are closely linked to RNA editing in plant organellar transcripts, as it is explained next.

RNA editing is a post-transcriptional modification to nuclear, plastid and mitochondrial genome-encoded transcripts, and occurs in a wide range of organisms [121,122,123]. In seed plants, 30 to 60 cytidine (C)-to-uridine (U) RNA editing sites are found in plastids and 300 to 600 sites are found in mitochondria [122]. To date, more than 150 PLS-subfamily proteins, including the DYW type, have been identified as editing site-recognition factors [123,124,125]. The binding of these PPR proteins to short *cis*-elements immediately upstream of editing sites is required for C-to-U processing [123].

In Physcomitrium, RNA editing occurs at only two sites in chloroplasts [126] and at 11 sites in mitochondria [69,127]. Among these 13 C-to-U editing sites, only three editing events at the *ccmFc*-C103, *ccmFc*-C122 and *nad5*-C598 sites also occur in Arabidopsis mitochondria. Nine out of ten Physcomitrium DYW-type proteins were identified as editing site-recognition factors [54,58,64,65,69,71,72], indicating that each DYW-type protein participates in one or two editing events (Table 1). This is the first full assignment of DYW-type editing protein factors to all their organellar editing sites in a plant species. Interestingly, the moss, *Funaria hygrometrica*, a closely related species of Physcomitrium, lacks both the PpPPR_56 ortholog and its target *nad3*-C230 and *nad4*-C272 editing sites. *F*. *hygrometrica* has nine DYW-type proteins but lacks the PpPPR_56 ortholog [65]. This suggests that DYW-type genes and their cognate editing sites were mutually constrained during their evolution [65,128].

In Physcomitrium chloroplasts, C-to-U RNA editing at the *rps14**-C2* site occurs at a high efficiency (80%) and creates a translation initiation codon AUG. In addition, the *rps14*-1C site in the 5′-UTR is edited at a low efficiency (5%) [129]. These editing sites also exist in the related moss, *F*. *hygrometrica*, but are not found in the chloroplasts of higher plants. The knockdown of the *PpPPR_45* gene reduced the extent of RNA editing at the *rps14*-C2 site, whereas the over-expression of *PpPPR_45* increased the levels of RNA editing at both the *rps14*-C2 site and its neighboring -C1 site. This suggests that the expression level of PpPPR_45 may affect the extent of RNA editing at the two neighboring sites. The efficiency of RNA editing at the *rps14*-C2 site was 70–80% in the young protonemata and decreased to 20% in old protonemata and the fully developed leafy shoots [129]. Thus, the RNA editing of this site is regulated in a tissue- and stage-specific manner and it may affect the efficiency of *rps14* mRNA translation.

## 8. Molecular Basis of RNA Editing in Physcomitrium

Although DYW-type proteins play a role in RNA editing as site-recognition factors in land plants, including Physcomitrium, the role of the DYW domain in RNA editing has long been elusive. In 2007, Salone et al. [130] proposed a hypothesis that the DYW domains catalyze C-to-U RNA editing. This is because the DYW domain contains a conserved region that includes invariant residues that match the active site of cytidine deaminases (C/HxE….PCxxC) from various organisms. Moreover, there was a correlation between the presence of nuclear DYW genes and the occurrence of organelle RNA editing among land plants [130,131]. Recently, this hypothesis was proved by in vivo orthogonal RNA editing assays in *Escherichia coli* and in vitro assays with purified proteins from Physcomitrium DYW-type proteins [132,133]. The crystal structure of the DYW domain of Arabidopsis OTP86 was recently determined, showing the potential RNA path on the DYW domain and identifying key residues required for the regulation and catalysis to occur [134]. Thus, a repeated PLS motifs-tract recognizes an immediate upstream sequence from a target editing site and a DYW domain catalyzes the C-to-U RNA editing reaction.

The chloroplast ribonucleoprotein (cpRNP) family containing two RRMs associates with large transcript pools and influences multiple plastid RNA processing steps [135,136,137,138]. In particular, cpRNPs also are involved in RNA editing in tobacco [139] and Arabidopsis [140]. CP31A, a member of the cpRNP family, influenced the efficiency of editing at 13 sites in Arabidopsis chloroplasts [140]. In Physcomitrium, cpRNP-like proteins, PpRBP2a and PpRBP2b, are present in the chloroplasts [141,142]. KO mutants of either one or two *PpRBP2* genes exhibited a WT-like phenotype and the efficiency of RNA editing at the *rps14* sites was not altered in the KO mutants. This suggests that PpRBP2a and 2b are functionally distinct from Arabidopsis cpRNPs and might not be required for RNA editing in mosses. Organellar RRM-containing protein, ORRM1, was reported to be required for plastid RNA editing at multiple sites in Arabidopsis and maize [143] but is not found in Physcomitrium.

Several other protein factors that are involved in organellar RNA editing were identified in flowering plants. Multiple organellar RNA-editing factors (MORFs)/RNA editing interacting proteins (RIPs) are required for plastid and mitochondrial RNA editing in flowering plants [144,145]. Ten members of the MORF/RIP family were identified in Arabidopsis. They interacted with each other and also with some PPR editing factors and formed specific homo- and heteromeric interactions [146]. These factors are organized in a higher ordered editing complex (~200 kDa, called the “editosome”) [145]. The protein components of the editosome vary depending on each target site in either plastids or mitochondria. Two additional proteins, protoporphrinogen IX oxidase 1 (PPO1) and organelle zinc finger (OZ), were also characterized as general editing factors [147,148]. Notably, PPO1, a critical enzyme for the tetrapyrrole biosynthetic pathway, plays an unexpected role in chloroplast editing at multiple sites in Arabidopsis [147]. PPO1 interacts with three chloroplast MORF proteins but not with the PPR proteins, suggesting that PPO1 controls the level of chloroplast editing via the stabilization of the MORFs. The OZ family contains four members, OZ1–4, in Arabidopsis and the disruption of *OZ1* led to an alteration in the level of editing of most sites in chloroplasts [148]. OZ1 interacts with PPR editing factors and ORRM1, but not with MORFs. The OZ family is present in many plant lineages but not in algae. Physcomitrium and Selaginella encode OZ-like proteins but they have minimal similarity. A new imprinted gene, *NUWA* (At3g49240), encoding a P-type PPR protein has influenced RNA editing in plastids and mitochondria [149,150]. NUWA enhanced the interaction of E+-type PPR protein and DYW2, a short, atypical DYW protein [150]. The overexpression of cationic peroxidase 3 (OCP3) also affected the editing of multiple sites in the chloroplast *ndhB* transcript [151]. Porphobilinogen deaminase HEMC interacted with the PPR protein, AtECB2, to achieve chloroplast RNA editing [152]. Curiously, most of these RNA editing-related proteins are not found in Physcomitrium, suggesting that the RNA editing machinery is largely different between mosses and seed plants. At present, DYW-type PPR proteins are a sole key player required for RNA editing in Physcomitrium. In contrast, in Arabidopsis, four types of PPR proteins are involved in C-to-U RNA editing: DYW type (many cases), E+ type (CRR4, SLO2 and CWM1, etc.), P type (NUWA) and a short, atypical DYW protein (DYW1, DYW2 and MEF8, etc.) [153]. RNA editing occurs in the editosome complex, and it is specific to a respective editing site (Figure 4). However, the possibility that unidentified factors are necessary for site-recognition, and the efficiency of RNA editing events together with DYW-type proteins in Physcomitrium cannot be excluded. Unlike the complex editosome of seed plants, RNA editing may occur in a simpler editing complex that is composed of a single DYW-type PPR editing protein and a few other unidentified non-PPR editing factors, at least in mosses (Figure 4).

## 9. Additional Function of the DYW Domain

Although most DYW-type proteins are responsible for RNA editing, some are involved in RNA splicing [53] or RNA cleavage [21]. The DYW-type PpPPR_43 protein is required for the group II intron splicing of the mitochondrial *cox1* transcript [53] (Figure 3). Its DYW domain is distinct from the other nine DYW domains of Physcomitrium proteins. Arabidopsis DYW-type CRR2 is required for the intergenic RNA cleavage of plastid *rps7*-*ndhB* pre-mRNA [21]. The DYW domain of CRR2 was indispensable for the cleavage of the target RNA in vivo [154]. The DYW domain of OTP85 (At2g02980) possessed endoribonuclease activity in vitro [155]. Its DYW domain contains the cytochrome *c* family heme-binding site signature (CxxCH), which overlaps with the active site of cytidine deaminase. The mutation of this signature to GxxGH resulted in a significant reduction in RNA cleavage activity. This suggests that the CxxCH motif is required for endoribonuclease activity of the DYW domain. The DYW domains of PpPPR_56, 71 and 77 proteins also showed endoribonuclease activity in vitro [116]. Thus, Physcomitrium DYW-type proteins are involved in RNA editing but may also function in certain RNA processing events in organelles. This possibility needs to be further investigated.

## 10. Conclusions and Future Perspective

As described above, the size and constitution of the PPR protein family are largely different between Physcomitrium and flowering plants. Some PPR proteins such as PpPPR_66/AtPPR66L and PpPPR_63, 67,104/AtPRORPs show the same or similar function, but some proteins including pTAC2 and its homolog, PpPPR_64, have a diverse function in Physcomitrium and Arabidopsis. Mitochondrial PpPPR_9, 31 and 43 are PPR splicing factors that are specific to Physcomitrium because their target introns are present in this moss but are not in Arabidopsis. There are only 13 RNA editing sites in Physcomitrium, but there are over 500 in Arabidopsis. At least three types of PPR proteins (E+, DYW and P type) are involved in RNA editing in Arabidopsis while only the DYW-type proteins are required for editing in Physcomitrium. Of the three editing events that are conserved in both plants, editing at the mitochondrial *nad5*-C598 site requires DYW-type PpPPR_79 in Physcomitrium but it needs the tripartite CWM1 (E+ type), NUWA (P type) and DYW2 (atypical DYW type) in Arabidopsis. Several PPR proteins are found in the plastid nucleoids [47,62] or mitochondrial ribosomes of Arabidopsis [156]. In contrast, at present, no PPR proteins in nucleoids or ribosomes have been identified in Physcomitrium. The PPR protein, GUN1, is known to be required for chloroplast-to-nucleus retrograde signaling [70]. Although GUN1 is an ancient protein that evolved within the streptophyte algal ancestors of land plants, but it has no role in chloroplast retrograde signaling in the streptophyte alga, *Coleochate orbicularis,* or in the liverwort, *M. polymorpha* [157]. Its role in chloroplast retrograde signaling probably evolved more recently. Thus, the evolutionarily conserved PPR proteins are not always functional orthologs and their function may have been expanded and diversified during plant evolution.

The proplastids of seed plants differentiate to various types of plastids including etioplasts, chloroplasts and chromoplasts. However, such plastid differentiation is not observed in Physcomitrium, and the moss gametophytes always contain chloroplasts that develop even while they are in the dark [158,159]. Therefore, it is considered that plastid gene expression is differentially regulated in a plastid-type specific manner in seed plants, although this is not likely in Physcomitrium. This may imply that the post-transcriptional regulation in plastids is more complex in seed plants than it is in mosses. Alternatively, seed plants may require more PPR proteins to accomplish the post-transcriptional regulation of gene expression specific to each plastid type. In contrast, mosses, including Physcomitrium, might retain a minimum set of PPR proteins that are required for the post-transcriptional regulation of plant organellar gene expression. To understand the basal and molecular mechanism of post-transcriptional regulation in plastid and mitochondrial gene expression, further identification of all the target RNA molecules recognized by the Physcomitrium PPR proteins and a characterization of their functions needs to be urgently performed. This will provide clues to identify the primary (primordial) function of PPR proteins in land plant lineages.

## Figures and Tables

**Figure 1 plants-11-02279-f001:**
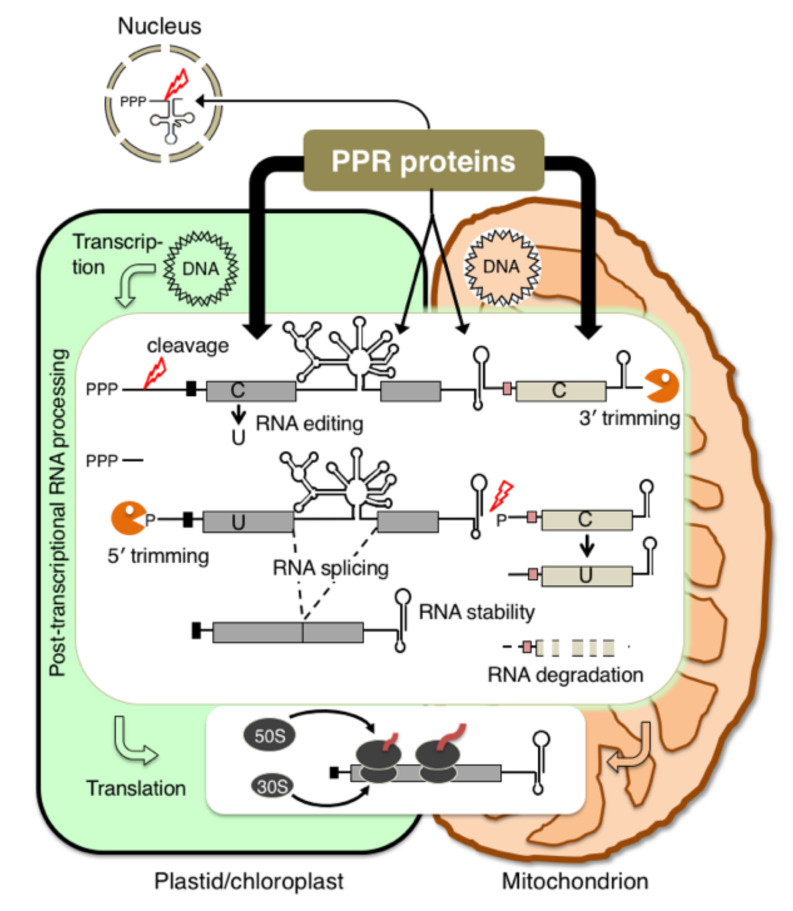
The complexity of post-transcriptional RNA processing events in plastids/chloroplasts and plant mitochondria. Plastid/chloroplast and plant mitochondrial genes are often transcribed as long polycistronic RNA precursors. Several post-transcriptional RNA processing events are necessary to produce the mature mRNAs: 5′ end processing by endonucleolytic cleavage, 5′ and 3′ ends of pre-mRNAs are trimmed by exonucleases (shown by the Pacman icons), specific cytidines (C) are edited to uridines (U), introns are spliced, and intercistronic regions are cleaved. Translation initiation is also an important step in the regulation of plant organellar gene expression. All these steps proceed via the participation of numerous nucleus-encoded RNA-binding PPR proteins. Almost all PPR proteins are imported into either plastids/chloroplasts or mitochondria and some members are dually targeted to both organelles. Small members of PPR proteins are imported to the nucleus, where they are involved in pre-tRNA maturation.

**Figure 2 plants-11-02279-f002:**
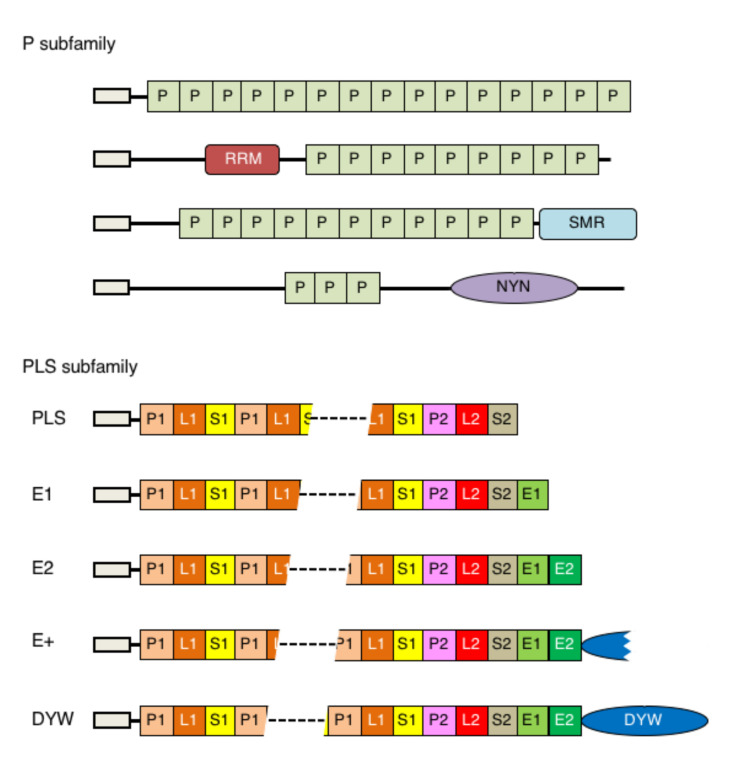
Architecture of pentatricopeptide repeat (PPR) proteins in land plants. **S**chematic structures of PPR proteins in different subfamilies and types are shown, according to Cheng et al. [15]. N-terminal boxes indicate a transit peptide targeting to organelles. The number of PPR motifs in each protein varies from two to more than thirty, and the first motif can be any of P, P1, L1 or S1. The P subfamily consists of P motifs only or P motifs and additional functional domain(s) such as RRM, SMR or NYN, as is described in the text. The E+ type consists of proteins with a degenerate or truncated DYW domain. The PLS subfamily is composed of 10 to 28 repeated motifs in Physcomitrium.

**Figure 3 plants-11-02279-f003:**
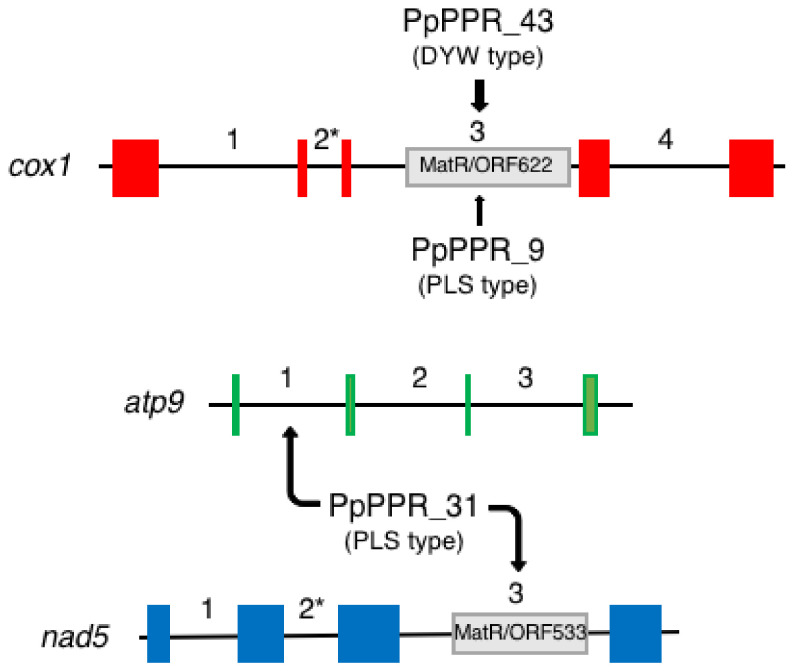
PPR splicing factors and their intron targets in Physcomitrium. **T**he *cox1* gene is interrupted by four introns and the *atp9* and *nad5* genes by three introns in Physcomitrium. The second introns (marked with asterisks) of *cox1* and *nad5* genes are group I introns. The other introns are group II introns. MatR/ORF622 or ORF533 indicates an intron-encoded maturase-like protein. PpPPR_31 is involved in the splicing of the first intron of *atp9* and the third intron of *nad5*. The splicing of the third intron of *cox1* requires two PPR proteins, PpPPR_43 and PpPPR_31. PpPPR_43 may be a major factor in the splicing of *cox1* intron 3, and PpPPR_9 may assist its splicing as an auxiliary factor.

**Figure 4 plants-11-02279-f004:**
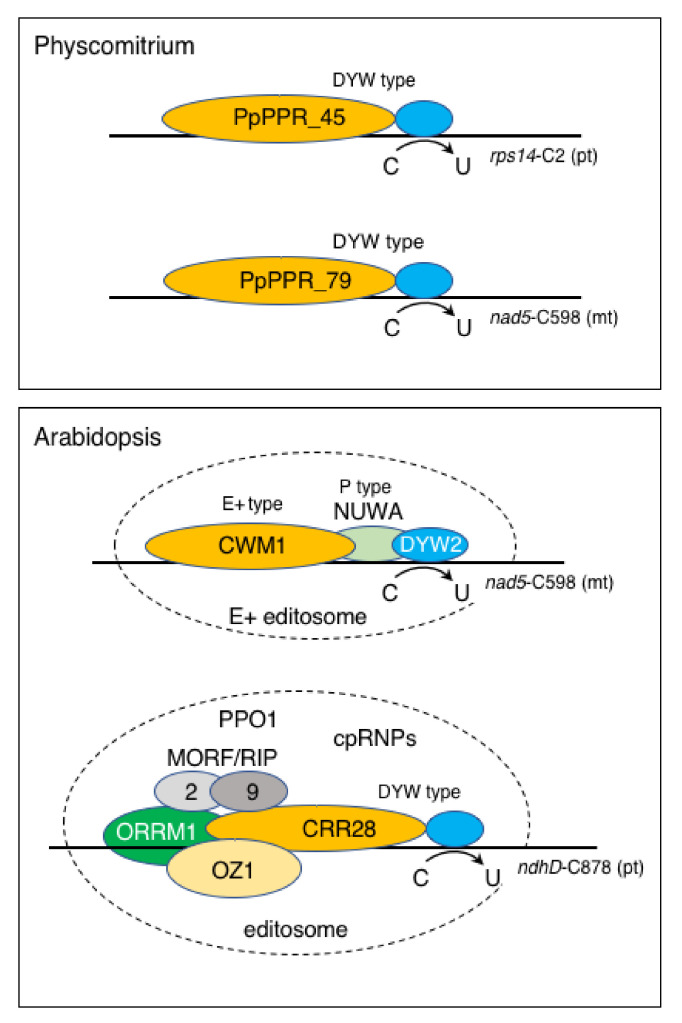
Models of RNA editing machinery in Physcomitrium and Arabidopsis. In Physcomitrium, nine DYW-type PPR proteins are responsible for editing at all 13 sites. They are a sole key player required for RNA editing. A PLS repeats-tract recognizes the upstream sequence of target site(s) and its DYW domain catalyzes C-to-U editing. Editosomes, including the non-PPR proteins involved in editing, have not yet been identified in Physcomitrium. In Arabidopsis, DYW- and E+-type PPR proteins are responsible for editing-site recognition. Most E+ type proteins function *in trans* with a short, atypical DYW2 protein for editing. Examples of an RNA editing event, the PPR and the non-PPR protein components required for editing at *rps14*-C2 (pt), *nad5*-C598 (mt), *ndhD*-C878(pt) sites, are illustrated. pt and mt in parentheses indicate plastid and mitochondrial editing sites, respectively. Members of non-PPR families (MORF/RIP, ORRM and OZ) are partially redundant. The editosome requires multiple copies of non-PPR factors. Most plastid and mitochondrial editosomes usually contain multiple non-PPR proteins in Arabidopsis but not in Physcomitrium. Models of Arabidopsis editosomes were modified from Andreés-Colaás et al. [150] and Sun et al. [148]. In the early land plants (mosses), the non-PPR editing factors that were identified in Arabidopsis were not encoded in their nuclear genomes. Unlike the complex editosome of seed plants, RNA editing may occur in a simpler editing complex, composed of a single DYW-type PPR editing protein and a few other unidentified non-PPR editing factors, at least in mosses.

**Table 2 plants-11-02279-t002:** Paralogous pairs of Physcomitrium PPR proteins.

Protein Name	Type	Additional Domain	Amino Acid Length (aa)	Amino Acid Identity (%)	Arabidopsis Homolog
PpPPR_3	P	RRM	939	62	At5g04810 (AtPPR4)
PpPPR_76			947		
PpPPR_5	P		611	79.2	
PpPPR_88			611		
PpPPR_6	P		451	62.5	
PpPPR_102			395		
PpPPR_7	P	LAGLIDADG	1195	71.5	At2g15820 (OTP51)
PpPPR_22			883		
PpPPR_13	P		657	75.4	
PpPPR_101			670		
PpPPR_16	P		637	70.8	
PpPPR_89			630		
PpPPR_17	P		489	58.5	At4g39620 (AtPPR5)
PpPPR_80			482		
PpPPR_19	P		958	65.8	At4g34830 (MRL1)
PpPPR_51			982		
PpPPR_26	P		1110	71.2	
PpPPR_40			961		
PpPPR_27	P		487	68.2	At3g53170
PpPPR_35			532		
PpPPR_39	P		582	58.3	At3g42630 (PBF2)
PpPPR_73			603		
PpPPR_42	P	SMR	936	56.1	At5g02830
PpPPR_59			935		
PpPPR_58	P		530	74.2	At4g35850
PpPPR_61			522		
PpPPR_66	P		578	77.3	At2g35130 (AtPPR66L)
PpPPR_72			578		
PpPPR_63	P	NYN	655	60.7–79.1	At2g32230 (PRORP1), At2g16650 (PRORP2), At4g21900 (PRORP3)
PpPPR_67			791		
PpPPR_104			993		
PpPPR_75	P	SMR	871	91.5	At2g31400 (GUN1)
PpPPR_85			871		
PpPPR_82	P		717	76.8	
PpPPR_84			717		
PpPPR_92	P		1010	73.7	At4g30825 (BFA2)
PpPPR_94			1000		

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
