# Peer review of "An Overview of Pentatricopeptide Repeat (PPR) Proteins in the Moss Physcomitrium patens and Their Role in Organellar Gene Expression"

_plants, 2022, doi:10.3390/plants11172279_

Round 1

Reviewer 1 Report

The review article entitled “An overview of pentatricopeptide repeat (PPR) proteins in the moss Physcomitrium patens and their role in organellar gene expression”, utilized 

Background: To elucidate the functions of pentatricopepetide repeat (PPR) proteins, a reverse-genetics approach has been applied to Physcomitrium patens. To date, 22 of 107 PPR proteins were identified as essential factors required for either mRNA processing and stabilization, RNA splicing, or RNA editing. This review discussed the P. patens PPR gene family and their current functional characterization, focusing on similarity and diversity of functions of PPR proteins between and P. patens and flowering plants. 

This is an important review article, which provides insights into the PPR gene family in Physcomitrium and the functions of PPR. I recommend accepting the current manuscript. 

Author Response

Response to Reviewer 1 Comments

The review article entitled “An overview of pentatricopeptide repeat (PPR) proteins in the moss Physcomitrium patens and their role in organellar gene expression”, utilized 

Background: To elucidate the functions of pentatricopepetide repeat (PPR) proteins, a reverse-genetics approach has been applied to Physcomitrium patens. To date, 22 of 107 PPR proteins were identified as essential factors required for either mRNA processing and stabilization, RNA splicing, or RNA editing. This review discussed the P. patens PPR gene family and their current functional characterization, focusing on similarity and diversity of functions of PPR proteins between and P. patens and flowering plants.

This is an important review article, which provides insights into the PPR gene family in Physcomitrium and the functions of PPR. I recommend accepting the current manuscript.

Response:

I really appreciate your positive comment. I hope that you will find the revised version suitable for publication.

Reviewer 2 Report

This manuscript reviewed the progress in PPR gene family member, localization and function in moss Physcomitrium patens, and compared the similarity and diversity of functions of PPR proteins between P. patens and Arabidopsis, rice, maize and other plants. The author’s laboratory has focused on the study of P. patens PPR genes for many years and the manuscript is well organized. The main concern is that the tables and figures were not showed in the manuscript, and the author should provide these tables and figures in the revised manuscript. 

Author Response

Response to Reviewer 2 Comments

This manuscript reviewed the progress in PPR gene family member, localization and function in moss Physcomitrium patens, and compared the similarity and diversity of functions of PPR proteins between P. patens and Arabidopsis,rice, maize and other plants. The author’s laboratory has focused on the study of P. patens PPR genes for many years and the manuscript is well organized. The main concern is that the tables and figures were not showed in the manuscript, and the author should provide these tables and figures in the revised manuscript.

Response:

I appreciate your positive comment. I am really sorry that the tables and figures were not shown in the manuscript. Although all files of tables and figures were separately uploaded in the system for submission, they may have not been conveyed to the reviewers. In the revised manuscript, all figures and tables have been inserted in the main text.

Reviewer 3 Report

Typos

pp. 1

Name of author: Delete ‘and’.

pp. 3

Line 3 of the paragraph 2 in Section 3: ‘in’?

pp. 6

Line 2 of the paragraph 2 in Section 5.2.1: Change gothic letters into serif font.

pp. 7

Section 5.3: Change gothic letters into serif font.

Section 5.4: Change gothic letters into serif font.

pp. 8

Section 6: Change gothic letters into serif font.

pp. 9

Section 7: Change gothic letters into serif font.

Section 7: Change en dash into hyphen.

Author Response

Response to Reviewer 3 Comments

Typos

  1. 1

Name of author: Delete ‘and’.

  1. 3

Line 3 of the paragraph 2 in Section 3: ‘in’?

  1. 6

Line 2 of the paragraph 2 in Section 5.2.1: Change gothic letters into serif font.

  1. 7

Section 5.3: Change gothic letters into serif font.

Section 5.4: Change gothic letters into serif font.

  1. 8

Section 6: Change gothic letters into serif font.

  1. 9

Section 7: Change gothic letters into serif font.

Section 7: Change en dash into hyphen.

Response:

Thank you for the comments on typos. As pointed out, most of the typos have been changed in the revised version.